# How to support a co-creative research approach in order to foster impact. The development of a Co-creation Impact Compass for healthcare researchers

Anneke van Dijk-de Vries[1]*, Anita Stevens[2], Trudy van der Weijden[1], Anna J. H. M. Beurskens[1]

1 Department of Family Medicine, CAPHRI Care and Public Health Research Institute, Maastricht University, Maastricht, The Netherlands, 2 Research Centre for Autonomy and Participation of Persons with a Chronic Illness, Zuyd University of Applied Sciences, Heerlen, The Netherlands

☉ These authors contributed equally to this work.
* anneke.vandijk@maastrichtuniversity.nl

**Data Availability Statement:** All relevant data are within the paper and its Supporting Information files.

## Abstract

Active participation of stakeholders in health research practice is important to generate societal impact of outcomes, as innovations will more likely be implemented and disseminated in clinical practice. To foster a co-creative process, numerous frameworks and tools are available. As they originate from different professions, it is not evident that health researchers are aware of these tools, or able to select and use them in a meaningful way. This article describes the bottom-up development process of a compass and presents the final outcome. This Co-creation Impact Compass combines a well-known business model with tools from design thinking that promote active participation by all relevant stakeholders. It aims to support healthcare researchers to select helpful and valid co-creation tools for the right purpose and at the right moment. Using the Co-creation Impact Compass might increase the researchers' understanding of the value of co-creation, and it provides help to engage stakeholders in all phases of a research project.

## Introduction

Research evidence plays a major role in guiding how to provide healthcare in an effective, efficient, safe, and patient-centered way. However, the gap between research and clinical practice remains a continuing challenge [1]. As building partnerships between scientific researchers and non-academics has been considered as a way to tackle this issue, researchers are conducting more and more action research approaches like community-based participatory research [2]. These approaches follow the principles of co-creation by means of involving stakeholders as full and equal partners in all phases of the research process. Co-creation would lead to outcomes that are more likely to be acceptable, valuable, and enduring than traditional research approaches [3]. It has been associated with significant societal impact [2] through better implementation and utilization of products and services [4].

**Funding:** The study was embedded within LIME, a four-year innovation program that was funded by the Province of Limburg, Zuyd University of Applied Sciences, and Maastricht University. The Province of Limburg had no role in study design, data collection and analysis, decision to publish, or preparation of the manuscript.

**Competing interests:** The authors have declared that no competing interests exist.

Co-creation is a wide construct that has become popular in the world of designers, business, health service research, and the public sector [5]. In this article, co-creation refers to an open, active, and creative process in which all relevant stakeholders are engaged in an innovation process [2, 6–8]. It concerns "active and committed decision-making about a meaningful problem through respectful interactions and dialog where everyone's voice is considered" [9].

Due to several challenges, using successful co-creation in health research is complex. Embracing the principles of co-creation means that researchers face much more uncertainty than when they start with a clear perspective on the desired solution of the problem and follow a predetermined plan for data collection and analysis. Their role shifts from a fully in charge decision-maker toward an equal distribution of power between the research team and other, non-academic, stakeholders. This new role needs to be accepted from the start of the research process [10]. The voice of the end-users (i.e. patients and/or health professionals) is crucial to guiding the development of innovations. This means that the researcher has to collaborate with patients and/or healthcare professionals to get a thorough understanding of their problem analysis. Otherwise, innovations may not be compatible with the end-users' needs, values, contexts, and norms, and may hamper successful implementation in daily practice [11]. However, it is challenging to bring researchers, patients, and healthcare professionals together in face-to-face meetings and facilitate co-creation. Another challenge refers to researchers' primary commitment to conducting and finishing the study rather than focusing on realizing a sustainable impact afterwards. Attention to implementation, dissemination, and market introduction often appears too late in the process. Therefore, less time is available for defining a good value proposition and identifying barriers for implementation in daily practice [2]. Last but not least, successful co-creation is complex due to factors like communication, planning, expectation management, and time and money to put effort into meaningful collaboration with stakeholders [12].

In order to generate societal, economic, and scientific research impacts, it is crucial for researchers to find ways to foster a co-creation process with relevant stakeholders [2]. There are various tools, roadmaps, and principles regarding the involvement of patients, health professionals, or other stakeholders, and to stimulate the process of co-creation [2, 10, 13–16]. For example, the Design Thinking approach offers helpful creative tools to engage with end-users [17]. Business and management literature also provide insights into co-creating value and thinking about revenue models [18]. However, although numerous value co-creation frameworks and tools from different professions are available on the internet, it is complicated for healthcare researchers to gain insight into when to use what tool for what purpose. They need tools to engage stakeholders in various steps of their research, from idea generation to implementation and dissemination. Therefore, it is not surprising that the literature shows a diversity of practices in the ways in which researchers engage with stakeholders [19]. This raises the question: How can healthcare researchers be supported to select helpful and valid co-creation tools for the right purpose and at the right moment, to foster their research impact? This article describes the bottom-up development process of an instrument and presents the final outcomes, which is a compass to support co-creation in research practice.

## Methods

### Setting

The study is embedded in the Dutch innovation program "Limburg Meet" [in English: Limburg measures] (LIME), a network of researchers, healthcare professionals, entrepreneurs, educational institutes, and citizens/patients/clients in the southern part of the Netherlands. The program aims to develop, redesign, and implement smarter measurement methods and

more efficient data collection for better healthcare. To promote value co-creation between all stakeholders, a co-creation team was set up to support the co-creative approach in the LIME projects. LIME has four main research themes, which are quality of care measurement (three projects), personalized wearables (two projects), Point-of-Care tests (one project), and data management systems (one project). These projects are conducted by independent research teams. Most of these researchers have a scientific educational background in social sciences, health sciences, psychology, or a more biomedical background like biophysics. The co-creation team, including two health researchers (two authors of this article: AD, AS) and one care technology innovation expert developed knowledge and facilitated practical support for co-creation to these researchers.

## Design

A participatory action research design was applied to involve researchers in an iterative and interactive approach in order to develop an instrument that would help them improve their research [20]. The key features of our systematic development process included a needs assessment, design of a prototype, "alpha" testing, "beta" testing, and the production of a final version for further evaluation and dissemination [21]. The iterative development process is visualized in Fig 1.

**Phase 1: Needs assessment.** Phase 1 aimed to define a list of requirements for a guide to support co-creation within LIME. A needs assessment was carried out by members of the co-creation team (AD, AS). Semi-structured interviews (n = 21) were held with eight senior staff members (i.e. full professors and associate professors), seven postdoctoral researchers, and six PhD students. Their expertise was related to health services research, beta sciences and technology, and clinical data science.

We first interviewed project leaders to get to know more about their research plans and to ask about their needs for co-creation in their project. Several months after the launch of the LIME program, we interviewed junior researchers and/or their daily supervisors to gain more insight into their experiences with co-creation, and to look forward to their challenges and needs in the co-creation process. The interviews were conducted during five individual meetings and eight group meetings with members from the same project team. All interviews lasted between 30–60 minutes. We made field notes during the interviews and wrote minutes of the discussions. The interviews were audio recorded in order to make verbatim transcriptions. Researchers' needs regarding co-creation were also discussed during two quarterly LIME co-creation group meetings with respectively six and four project members of LIME. All these members also took part in the interviews, except for a PhD student in health services research, and a senior lecturer in economics and business. We analyzed the interviews and group discussions by direct content analysis [22]. The interview findings were translated into a *list of requirements* concerning the functional aspects and content of a guide for a co-creative research approach.

**Phase 2: Design.** Phase 2 aimed to make a first prototype. We started with a literature search to get an impression of available frameworks and tools for co-creation. It was an open exploration rather than an exhaustive systematic search. The search was guided by our list of requirements (Phase 1) and focused on frameworks and tools for the context of healthcare research. We searched electronic databases for original peer-reviewed manuscripts and consulted experts to include gray literature such as reports, websites, and manuals. The framework and tools were put together into Prototype I.

**Phase 3: Alpha testing.** The alpha testing phase aimed to evaluate Prototype I, involving five experts in the fields of 1) design thinking; 2) business and communication; 3)

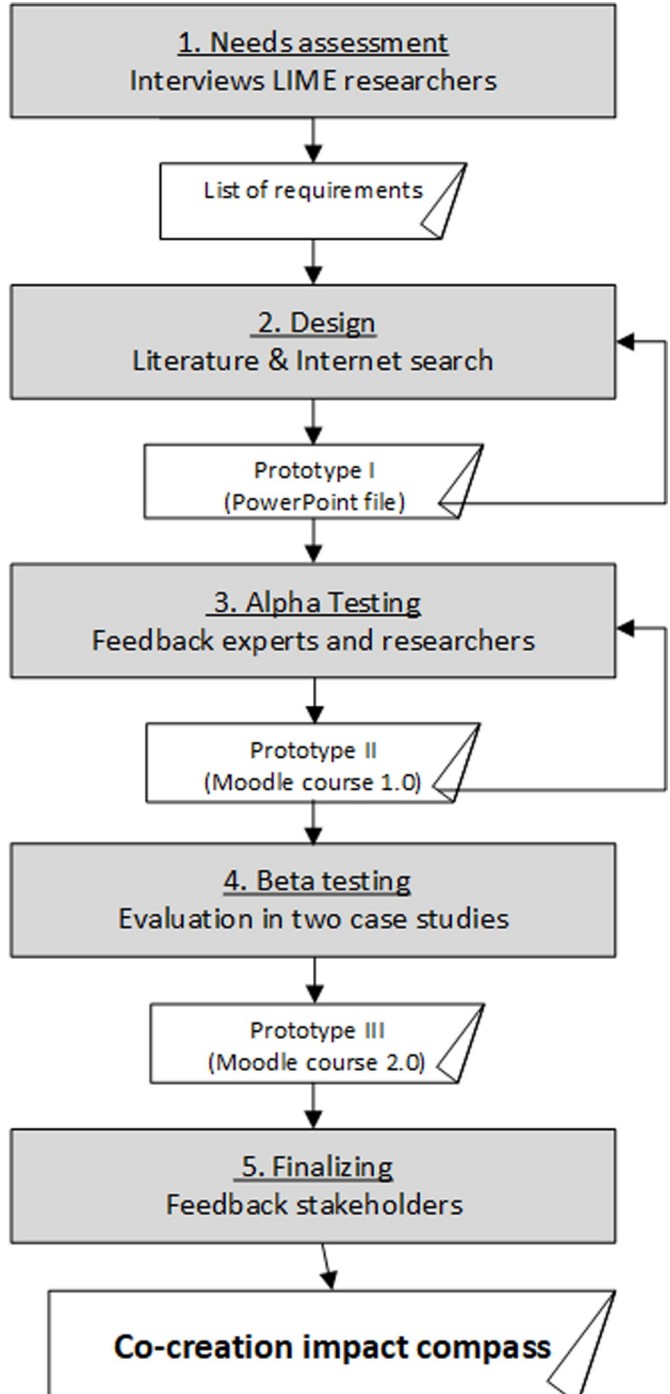

**Fig 1. Iterative development process.**

patient participation and citizen empowerment in healthcare research; 4) business development and innovation; and 5) business and management. Furthermore, LIME researchers, who also were involved in Phase 1 gave feedback in two group sessions of five and four participants, respectively. Participants were asked to give their first reaction and critical remarks regarding the usefulness of the co-creation guide. Usefulness refers to the degree

to which the guide serves the users' needs, and its applicability in the research practice context [23]. After each interview or group meeting, the feedback was discussed (AD, AS, AB) and directly used to revise and improve the prototype for the next iteration. The alpha testing phase concluded with Prototype II.

**Phase 4: Beta testing.** The beta testing phase aimed to evaluate Prototype II using researchers who were not involved in Phases 1–3 of the development process. We purposively selected two cases in which a health innovation was developed and evaluated. These cases are described in S1 File.

Case 1 concerned a project that was already finished, so we were able to reflect on the project's development, evaluation, and dissemination. In Case 2, the researcher had recently started an action-based research project. This enabled us to gain insight into the value of the guide for researchers in an early stage of their study. Both researchers were first asked to reflect on their co-creative approach and the perceived impact of their project. Then, the interviewer presented Prototype II and asked for reflection on terminology, look-and-feel, and comprehensibility, as well as the usefulness of the guide for their research practice. The interviews were audio recorded and field notes were made during the interviews. After the beta testing phase, the feedback was discussed and processed into Prototype III.

**Phase 5: Finalizing.** Phase 5 aimed to determine the final version and further steps for dissemination. Individuals from research and education, who had followed a masterclass about co-creation, gave feedback on the guide. The feedback was processed by the co-creation team. The final version is presented in this article.

## Ethical considerations

The development process of the compass does not fall under the scope of the Medical Research Involving Human Subjects Act (WMO) and ethical approval by an institutional review board was not sought. All interviews were undertaken to involve relevant stakeholders in the co-creative development process. Participants were no subjects to procedures nor required to follow rules of behaviours. We verbally asked participants for their permission to audio record the interviews and group meetings. The two external researchers gave their written informed consent due to the critical reflection on their research projects.

## Results

The results section presents the outcomes of the different phases as described in the methods.

## Phase 1: Needs assessment

Several needs regarding supporting a co-creative approach emerged during the interviews and group discussions. The results of the needs assessment are summarized into a list of six requirements (see Table 1).

1. Respondents' reflections about co-creation showed that their meaning of the concept of co-creation was more than just good collaboration between project partners. Creating a common understanding of the concept of co-creation came forward as a basic requirement.

   First, I thought it is a good way of working together. But I think, my gut feeling says that it's more than that. Co-creation is about people's self-experience, feeling, collaborating from different perspectives. Maybe it's more about finding one another during the process at the right moment. So you bring each other at the right moment to a higher level.

2. Interactions with stakeholders were described as valuable and a "must" to reach successful implementation of innovations in healthcare. Perspectives from stakeholders lead to crucial insights and changes. The following example illustrates an eye-opener during a stakeholder meeting at an early stage of a project. The researcher discovered that the needs of the people on the work floor did not match with the perspective of the researcher:

> We [research team] wanted to develop a new questionnaire. But this was a 'no go' for all stakeholders at different levels, from the work floor up to management. They told us: "Don't bother us with a new questionnaire; we'll shoot you, so to speak. We want to have something simple, like three smileys: green, orange and red. That is what you should think of."

It showed that the compass should create awareness of the value of co-creation in research in order to foster co-creation.

3. Researchers expressed their need for diverse methods of co-creation in order to support ongoing interactions with stakeholders. Some interviewees asked for support to facilitate meetings in which decisions had to be made concerning the future phases of their project:

> At the beginning of the project we thought about co-creation ourselves. . . .. But, if you really want to know, honestly, I don't know how we will continue. We are going to have some very interesting sessions, but I don't know how we will design them. Maybe you [the co-creation team] can help us.

For effective decision-making processes, differences in language, power, and interests between stakeholders need to be taken into account:

> If we know what we are going to deliver, than we have to call together our stakeholders: what are we going to do now? You cannot just put all people together in one room. Not everyone is in the same game. Business, hospitals, patients, they all are different. How can we consider everyone's interests? We have to develop together without competition.

4. The researchers realized that they often use traditional research methods to gain insight into the needs of potential end-users, like interviews, focus groups, and surveys. They highlighted that they were eager to learn, but did not know how and where to find other appropriate methods and tools:

**Table 1. List of requirements.**

|   | The compass has to... |
|---|---|
| 1 | create a common understanding of the co-creation concept |
| 2 | create awareness of the importance of co-creation in research |
| 3 | support a collaborative process with stakeholders in every project phase |
| 4 | lead to appropriate selection of co-creation tools |
| 5 | contain tools that are easy to apply, include practical instructions, and background information |
| 6 | be useful for training purposes concerning co-creation in research |

Once you did something one way, you persist in doing it this way, while it's also nice to try a new method. The question is whether you have to use the same method for the same question in another group, for instance for patients as well as general practitioners–and don't they think it's too childish? That's our need for support: specific methods for certain groups.

5. According to the researchers, the selected tools had to be easy to apply, include practical instructions and information about their validity, in order to publish in scientific journals.

6. Some senior researchers emphasized their challenges concerning how to equip PhD students regarding co-creation, and wanted the compass to be useful for training purposes.

## Phase 2: Design

Based on the needs assessment, our literature search focused on finding a practical framework, and selecting and organizing co-creative tools. We searched for valid tools that addressed the user requirements such as being easily applicable in research projects.

**Selecting a framework.** For an underlying framework for selecting co-creation tools, we considered various innovation models such as Lean Innovation and the Stage-Gate-Model [24, 25]. Some models have a more linear perspective on the stages of ideation, prototyping, testing, and implementation, while others are divergent–convergent models that assume integrated evaluation and selection of ideas and concepts, like the Double-Diamond model [25]. We found that most toolboxes for co-creation are organized according to the (design) phases of an innovation project, for example understanding, conceptualizing, and testing [13, 15]. However, since creating mutual interactions with stakeholders and decision-making processes relate to all phases in a research project, we started to look for a more overarching framework with a focus on the ultimate goal of co-creation, that is: creating societal, economic, and scientific impacts by striving toward outcomes that are acceptable, valuable, and enduring [3]. In this regard, the Business Model Canvas (BMC) came forward as a valuable, wide-spread conceptual framework [26]. It is a commonly used business framework in the healthcare sector [27] and consists of nine building blocks, representing what value is provided to who, how this is done, and with what consequences [26, 28]. See Fig 2 for the BMC.

In the BMC, the "value proposition" in the middle describes how the customer segments will value from the unique service and product. The right side of the framework focuses on the "customer segments" (i.e., the target group), what "channels" the value proposition will be delivered through, and on keeping solid "customer relationships". The left side of the framework includes the "key partners" that need to be obtained through networking to contribute to the success of the innovation, the "key resources" that allows to run the operations effectively, and the "key activities" that are essential to the success of the innovation. The "cost structure" (i.e. costs associated with the innovation), and "revenue streams" (i.e. how the value propositions will bring in revenues from each target audience) are positioned in the lower side of the framework. The BMC can be used as an iterative framework that can be applied in all phases of a project, from generating ideas to the implementation phase in which dissemination processes and revenue streams become more important.

**Selecting and organizing the tools.** The BMC provokes critical questions about the value proposition of a research project, the needs of end-users and about key partners that need to be involved to create sustainable value. The best way to answer these questions is in an interactive process with research team members, end-users, and key partners. Therefore, we assigned tools related to potential questions in specific building blocks of the BMC. For example, the

| Key Partners | Key Activities | Value Proposition | Customer Relationships | Customer Segments |
|---|---|---|---|---|
| | Key Resources | | Channels | |

| Cost Structure | Revenue Streams |
|---|---|
| | https://www.strategyzer.com/canvas |

**Fig 2. The business model canvas.** Reprinted from [26] under a CC BY license, with permission from Strategyzer AG, original copyright 2010.

building block "customer segments" will provoke the question: Who are the people that may benefit from our research? Besides traditional interview methods, there are various, more creative tools from design thinking that will give a deeper understanding of a subject (divergent), or build a common reality about a certain topic (convergent). Our selection process was concentrated on tools that were successfully applied within LIME and/or are shown to be effective in scientific literature. We collected practical instructions, pictures, references to manuals, and scientific articles in which the tools have been described or applied.

To organize the selected tools in a meaningful way, we merged tools with the same purpose and assigned a symbol. These symbols were placed within the BMC framework. The more generic tools, i.e. aimed at deliberating and making shared decisions, did not fit within a specific building block of the BMC; hence, they were put in an outer circle.

The design phase resulted in Prototype I: An interactive PowerPoint file in two parts: 1] Introduction about co-creation in healthcare; and 2] Co-creation tools plotted within the BMC.

## Phase 3: Alpha testing

The interviewed experts and researchers were positive about the BMC as the underlying framework. For the healthcare researchers, the BMC was a new model that helped them to gain a broad overview of a co-creative research approach, including participation of both end-users and key partners as stakeholders. Some experts reflected on the complexity of the healthcare setting with unique funding mechanisms, separated buyer–user decisions, a strong regulatory framework, fragmentation of stakeholders, and different reimbursement models.

Based on the alpha testing phase, the prototype needed some adjustments:

- A building block "customer segments" would not completely fit the complex healthcare system, in which different types of "customer segments" or "payers" in healthcare innovations exist, including patients, care providers, and health insurers. Therefore, the block "customer segments" was renamed into "target group" which can be differentiated in final end-users (e.g. patients) and providers of the innovation (e.g. caregivers). The building block "key partners" was renamed into "stakeholders".

- The division of the building blocks was reshuffled in order to highlight the need for active involvement of both "target group" and "stakeholders" for the process of value creation. Therefore, both building blocks were connected with an arrow.

- The block "revenue streams" was renamed into "societal, economic and scientific impact" since experts suggested that a predominant focus on a business model would not fit healthcare research, in which meaningful outcomes also relate to quality of life and care.

The framework with tools was named the "Co-creation Impact Compass" to highlight the direct connection between the value of co-creation and the aim of generating societal, economic, and scientific impacts. It has been shown in Fig 3. The figures in the framework symbolize six leading questions in the compass: 1) How do we understand the target

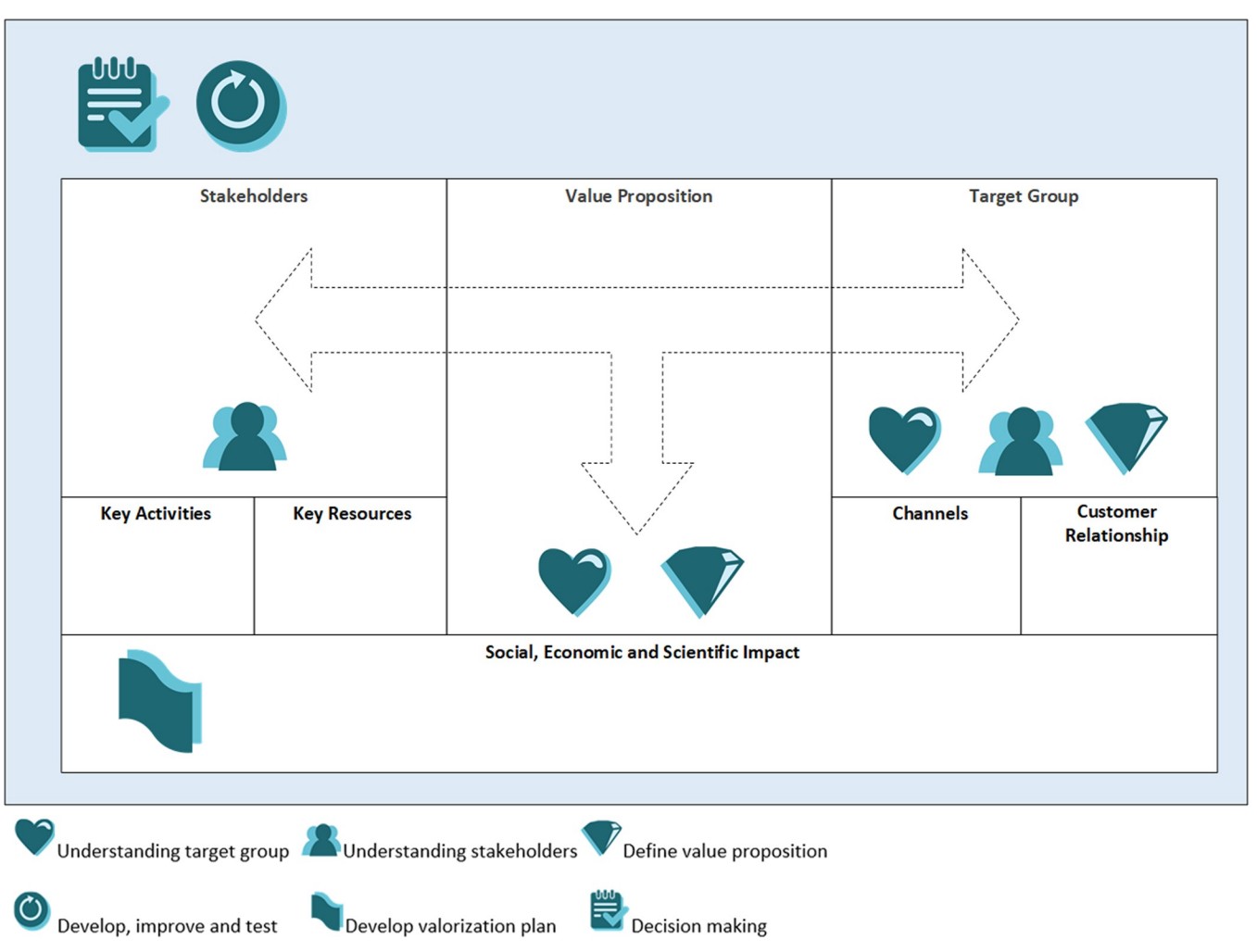

**Fig 3. The Co-creation Impact Compass.**

group?, 2) How do we get insight into our stakeholders?, 3) What is the value proposition?, 4) How do we develop, improve and test our product/intervention?, 5) How do we develop a valorization plan?, and 6) How do we collect information and make shared decisions with the stakeholders?

The PowerPoint version of the compass was embedded in an online learning program "Moodle" used by Zuyd University of Applied Sciences (Zuyd) to increase its accessibility and educational opportunities. The result of the alpha testing phase was Prototype II: The Co-creation Impact Compass embedded in Moodle course 1.0.

## Phase 4: Beta testing

The participating researchers retrospectively (Case 1) and prospectively (Case 2) reflected on their co-creative research approach, on the perceived impact of their project, and on the usefulness of Prototype II of the Co-creation Impact Compass for research practice. With regard to the framework, issues were raised concerning the building blocks, value proposition, target group, stakeholders, and impact.

**Value proposition.** Both respondents had started with a societal problem rather than the actual experiences of a specific target group. The framework and tools were perceived as helpful and challenging to force researchers to test and sharpen their value proposition:

> As a novice, you assume that everyone will understand the importance of your research. So, why should I have to create support for my research proposal? After a while, I realized: The people who are part of my research network all have a shared interest in improving interprofessional education and collaboration. Others need to be persuaded about the relevance of my project. . . (Case 2)

**Target group.** Defining the target group appeared to be complex. The researchers were challenged by the compass to define and check their perspective on "the target group." The intervention in Case Study 1 was developed for practice nurses. Retrospectively, the researcher realized that he also should have focused on general practitioners:

> Practice nurses always have to get approval from their general practitioner or practice manager. That is what I've learned during my research study. Maybe. . . general practitioners should have been our target group as well. (Case 1)

The interviews showed that a target group in terms of impact of the research (i.e. efficient patient care) could be different from a target group that is defined as "end-users of the intervention":

> A priori, I presumed that I do this research for patients. It is aimed at more integrated care. After six months, I realized that it would solve a societal problem by generating a more efficient and cheaper care approach. In addition to patients, the ultimate impact of my research is also on general practitioners and other healthcare professionals since they will become more capable to help patients. It decreases their workload. (Case 2)

Reflecting on these issues, Researcher 2 concluded that trainees in family medicine would be a primary target group:

The idea is to offer them [trainees in family medicine] several tools, several interventions, from which they can choose the most valuable. So, finally, my target group are trainees since they determine whether they will use the intervention. (Case 2)

**Stakeholders.**  Both case studies illustrate the relevance of becoming involved with the people and organizations in the community. It was recognized that the compass supports researchers to understand which key players are necessary for effective partnerships:

By coincidence, during a conversation with a practice nurse, it became clear to me that she was a kind of leader in a care group. She said: "When you have the whole care group on board, you immediately have 100 practices including general practitioners, practice nurses . . ." (..) If I could do it [my research] again, I would immediately focus on getting all care groups on board and roll out my activities from these care groups. (Case 1)

Besides the fact that researchers will gain insight into the importance of the partners' influence, needs, and interests, the compass was perceived as valuable for the involvement of members of the research team, sounding board, and steering committee:

The tools triggered me. I think, they could be very valuable for a new trainee. Where do you start, who is the target group, what to expect from project advisors (whom I did not know in the beginning). Probably, the project advisors could have been of greater benefit or support, when I had involved them in an earlier stage of the project. (Case 1)

**Impact.**  Both researchers realized that their study proposals did not include dissemination as a final attainment level of their research project. Primary concerns related to the development of the intervention and demonstrating its effectiveness. Questions about dissemination only became relevant at the end of Case 2:

Implementation issues were regarded as already naturally covered in our effectiveness research. It was all about measuring effectiveness. At a later stage in the project, we started to talk about: "We are reaching the end of the project, and now?" At that time, I discovered the Business Model Canvas, and questions about costs-effectiveness and sustainability became relevant. Issues regarding efficiency, the long-term picture, should be in your research proposal! (Case 1).

The perceived usefulness of the Co-creation Impact Compass and the researchers' suggestions for improvement are summarized in Table 2. Furthermore, they suggested that some guidance or training would be needed to use the compass.

Table 2.  Reflections of external researchers (n = 2) on the Co-creation Impact Compass.

|  | Perceived usefulness | Suggestions for improvement |
|---|---|---|
| Framework | • The added value of the framework is that it provokes relevant questions during the starting phase of a research project. | • Examples and definitions of the building blocks of the BMC should be added because healthcare researchers are not familiar with the terminology. |
| Co-creation tools | • The tools can contribute to an effective collaboration process within the research team and with stakeholders. • The tools give a broader view on potential research methods. | • The framework needs to be understood before the questions and tools become relevant. • A direct link between the research questions and the tools needs to be presented as a second pathway in the compass (Moodle). |

Phase 4 led to minimal changes of the compass. To meet the necessity of additional support or training in using the compass, we developed a masterclass "Co-creation" for educators from Zuyd. The masterclass consisted of two three-hour training sessions, with practical application of different co-creation tools that are included in the compass. The beta testing phase resulted in Prototype III: The Co-creation Impact Compass embedded in Moodle course 2.0.

## Phase 5: Finalizing

The web link of the Moodle course 2.0 was provided to all participants of the masterclass "Co-creation" for educators from Zuyd. Feedback was given during one individual meeting and one group session with four educators (design thinking, nursing, law, innovation in health-care). Overall, the educators were positive about the comprehensiveness of the compass, the clear instructions, the fancy symbols, the structured way of organizing the co-creation tools, and the opportunity for continuous actualization in the online application. Some were slightly confused since the chronological order of a project cannot be found in the underlying BMC framework. Suggestions for textual improvements in the explanation of the compass were given. Although the embedding of the compass within the Moodle course was seen as a first step to making it available for more people, the educators expressed their concerns and mentioned the limits of this online learning platform. They stated that additional instructions would be necessary to make the Moodle course usable without any additional support

**Table 3. Overview of co-creative tools included into the Co-creation Impact Compass.**

| Aim of the tool | Tool |
|---|---|
| *We want to understand our target group by….* | |
| Getting insight into latent and tacit knowledge of users. | Context mapping |
| Getting insight into detailed data about real-life activities. | Shadowing |
| Visualizing of a users' experience with a service or product. | Customer journey |
| Mapping one day in a life of users. | Day in the life |
| Mapping knowledge about behavior, thoughts and attitudes. | Empathy mapping |
| Developing archetypical descriptions of end-users. | Personas |
| *We want to get insight into our stakeholders by…* | |
| Identifying and analyzing relevant stakeholders. | Stakeholder mapping |
| Discussing roles regarding participation and decision-making. | Participation game |
| Discussing shared goal, expectations, contributions and struggles. | Value Pursuit |
| *We want to define the value proposition by….* | |
| Discussing an intervention in relation to the users' values and needs. | Value Proposition Canvas |
| *We want to develop, improve and test our innovation by…* | |
| Testing prototypes in different ways. | Prototyping |
| Testing the usability of prototypes in different ways. | Usability testing |
| Identifying usability problems in a user interface design. | Heuristic evaluation |
| *We want to develop a valorization plan by…* | |
| Setting up a valorization plan or business plan. | Business Model Canvas |
| *We want to collect information and make shared decisions by…* | |
| Visualizing systems and processes. | Process Mapping |
| Getting input from different perspectives. | World café |
| Reflecting on specific issues from different perspectives. | Thinking Hats |
| Prioritizing to select the most valuable ideas. | 100-dollar method |
| Reaching consensus | Nominal Groups Technique |

We processed the feedback into the definite version of our Co-creation Impact Compass, which has been presented in Fig 3. The compass is embedded in a Moodle course, an online learning program of Zuyd. Users need to click on a symbol to find tools that are useful to answer that particular question. Table 3 provides a list of all tools that are included in the Co-creation Impact Compass. S2 File provides a brief description of these tools, including references to scientific literature and gives more insight into how the tools can be applied for research purposes.

## Discussion

This article has been presented the development process and final version of the Co-creation Impact Compass. The study was aimed to support health researchers to select helpful and valid co-creation tools to foster their societal, economic, and scientific impacts. The Co-creation Impact Compass includes a modified business model as a framework, in which a selection of existing co-creation tools have been "framed". It increases researchers' understanding of the value of co-creation for societal, economic, and scientific impacts of their research project, and supports them to select helpful and valid co-creation tools for the right purpose and at the right moment. We selected tools that have already been used in scientific research practice.

The advantage of the Co-creation Impact Compass is its focus on both the "why" (doing research that matters) and "how" (tools for co-creation) of engaging stakeholders in scientific research practice. Most existing toolboxes are developed for one of these purposes, and are organized according to the different stages of a project. The framework of our compass provides an overview of all aspects that need to be addressed to generate a sustainable research impact. These aspects will provoke leading questions that lead to the co-creative tools.

To our knowledge, using a business model as an underlying framework for community-engaged healthcare research practice is new. The real essence, for both entrepreneurs and researchers, is creating value-capturing processes [29]. In research practice, it means that researchers need to communicate the value proposition in a clear and convincing way, and they must be aware of all stakeholders that influence the outcomes of their study. Using the BMC from the very beginning of a project as a framework forces them to focus on these aspects, and to outline how they will collaborate with stakeholders.

The iterative development process of the compass was based on a participatory action research design, with ongoing input from healthcare researchers and educators (end-users), and other stakeholders (experts from different fields of expertise). Due to the bottom-up approach, the content and usefulness of the compass were positively valued by the participants.

As most researchers are not educated in how to conduct patient and community involvement in research [30], the Co-creation Impact Compass seems valuable in postgraduate education at universities. A tailored training based on the compass has potential in research practice. It calls for more attention to a dissemination plan in an early stage of a project [2]. Although these activities will take extra time and effort in the early phases, they may increase the impact afterwards. Therefore, we recommend using the compass in the early phase of writing grant applications.

### Limitations

Although the compass is aimed at facilitating a co-creative research approach, it is not a guarantee for a successful co-creative process, or for a better outcome in terms of increased societal, economic, or scientific impacts. A range of internal and external project factors will determine the relevance and usability of knowledge [31]. If there are problems regarding issues like

governance and facilitation arrangements, the style of leadership, and how conflict is managed, co-creation efforts may fail [2].

We did not plot co-creative tools into all building blocks of the BMC. Decisions about the means, activities, and channels to reach end-users will result from generic tools that have been put in the outer space of the framework. Since this may be confusing and needs explanation for new users, it might diminish the face validity of the compass.

Our alpha and beta testing phases were mainly focused on the perceived value of the underlying framework of the compass, and not on the quality of the co-creation tools. More insight is needed with regard to the use of these tools in research. Moreover, the selection of the tools is not meant to be exhaustive as one might lose the overview. However, the content could be expanded in the future.

## Future directions

We intend to transfer the compass from Moodle into an open, online platform, and will use heuristic evaluation to identify usability problems in the user interface [32]. Another step is to gain more insight into experiences of researchers in using the compass in their practice, and their training needs. Further evaluation could also focus on the researchers' learning process. A co-creative approach may have implications for the motivation of researchers to work with patients and other stakeholders in the future [33]. Finally, it would be interesting to gain more insight into the effects of using the compass in terms of societal or economic impacts of a research project. This would further strengthen the evidence that a co-creative approach fosters the acceptability and societal value of research outcome [2, 3]. Studies that have examined the effectiveness of specific tools from our Co-creation Impact Compass, like personas [34] and consensus methods [35], also show this positive association between co-creation and impact.

## Conclusion

The Co-creation Impact Compass is a promising and useful guide to support researchers conducting a co-creative approach for research practice. The awareness of co-creation, the underlying business framework, and the leading questions toward appropriate co-creation tools support researchers in creating more impact.

## Supporting information

**S1 File. Description of the two case studies.**
(DOCX)

**S2 File. Overview of co-creative tools.**
(DOCX)

## Acknowledgments

We thank all researchers and experts who participated in the study. We thank John Rietman for his active contribution and guidance in how to apply several co-creation tools in practice.

## Author Contributions

**Conceptualization:** Anneke van Dijk-de Vries, Anita Stevens, Anna J. H. M. Beurskens.

**Data curation:** Anneke van Dijk-de Vries, Anita Stevens.

**Formal analysis:** Anneke van Dijk-de Vries, Anita Stevens.

**Funding acquisition:** Anna J. H. M. Beurskens.

**Investigation:** Anneke van Dijk-de Vries, Anita Stevens, Anna J. H. M. Beurskens.

**Methodology:** Anneke van Dijk-de Vries, Anita Stevens, Trudy van der Weijden, Anna J. H. M. Beurskens.

**Project administration:** Anneke van Dijk-de Vries, Anita Stevens, Anna J. H. M. Beurskens.

**Resources:** Anna J. H. M. Beurskens.

**Supervision:** Trudy van der Weijden, Anna J. H. M. Beurskens.

**Validation:** Anneke van Dijk-de Vries, Anita Stevens, Trudy van der Weijden, Anna J. H. M. Beurskens.

**Visualization:** Anneke van Dijk-de Vries, Anita Stevens, Trudy van der Weijden, Anna J. H. M. Beurskens.

**Writing – original draft:** Anneke van Dijk-de Vries, Anita Stevens.

**Writing – review & editing:** Anneke van Dijk-de Vries, Anita Stevens, Trudy van der Weijden, Anna J. H. M. Beurskens.

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
