## [Decision Letter · Decision Letter 0]

26 May 2020

PONE-D-20-05698

How to support a co-creative research approach in order to generate more impact. The development of a Co-creation Impact Compass for healthcare researchers

PLOS ONE

Dear Dr. van Dijk-de Vries,

Thank you for submitting your manuscript to PLOS ONE. After careful consideration, we feel that it has merit but does not fully meet PLOS ONE’s publication criteria as it currently stands. Therefore, we invite you to submit a revised version of the manuscript that addresses the points raised during the review process.

We look forward to receiving your revised manuscript.

Kind regards,

Paola Iannello

Academic Editor

PLOS ONE

Journal Requirements:

Additional Editor Comments (if provided):

Reviewers' comments:

Reviewer's Responses to Questions

**Comments to the Author**

1. Is the manuscript technically sound, and do the data support the conclusions?

Reviewer #1: Yes

Reviewer #2: No

2. Has the statistical analysis been performed appropriately and rigorously? 

Reviewer #1: Yes

Reviewer #2: N/A

3. Have the authors made all data underlying the findings in their manuscript fully available?

Reviewer #1: Yes

Reviewer #2: Yes

4. Is the manuscript presented in an intelligible fashion and written in standard English?

Reviewer #1: Yes

Reviewer #2: No

5. Review Comments to the Author

Reviewer #1: The main purpose of this study is to highlight how the co-creation process can be a functional method for health researchers by helping them involving all the stakeholders and improving the impact of their researches.

The paper is properly placed in the context of the previous literature by highlighting the potential of co-creation techniques but the lack of a reference framework for health care surveys. The study therefore aims to describe the process of creating a "compass" of co-creation tools and techniques that can be used by researchers in this field in an effective way (which tools/in which moment). However, in the “introduction” I recommend removing the word "demonstrates" from line 82 because this term refers to demonstrating the effectiveness of this tool, which is not the goal of this research, that is to DESCRIBE the creation of this tool.

Moreover, I think that the structure of some parts (especially of methods and results) should be modified in order to ensure consistency and linearity in the reading of the paper . I will resume all the proposed changes here:

1. I would move the "Setting" part to the beginning of the methods for introducing the framework in which the study fits, and specify which of the categories this study falls into.

2. I would remove the sub-paragraph "development process" and integrate it with the design. Too many sub-paragraphs continuously interrupt the reading. In this way, the design part is also linked to subsequent explanations of the various stages of the process.

3. In my opinion some information from the “results” section must be moved to the methods section (e.g. line 179 – 186; 286-290; 324-328 including the box describing the two cases) these are all details referring to the methodology and not to the results of the study), these would help the authors to better clarifies the methodology which is currently not very precise and, on the other hand, it streamlines the results.

4. I find the explanation of BMC unclear, which should be the central point of the paper. For example, what are the nine blocks? What do the nine blocks represent? How are they implemented in the compass?

5. Line 320: personally, I would find a way to put the symbol legend inside the image. Putting the legend inside the article breaks the reading. Rather, in the text, I would put a brief explanation of what you mean by each category.

6. Discussion: look above for the use of the word “demonstrates”

7. As concern line 45-456 since it explains what has been done in this study, I would move them before explaining the advantages (before line 438).

8. I suggest to add a “future direction” section and to merge here what you think are the future implications thanks to the use of this tool

9. In the conclusion section I suggest to highlight better what you mean with “impact”

Reviewer #2: Authors addressed a key topic in interdisciplinary applied research: engagement of all stakeholders and researchers in a specific field, that is, healthcare.

At the beginning of Authors' work, they stated that the aim is to present a "compass" to facilitate such engagement. However, it is not clear what this "compass" is: a tool; a method; a process of intervention? This is the first main weakness of this work: the goal and the nature of their work is not stated clearly.

This obfuscation affects all the other parts of this manuscript:

(i) A clear overview of existing methods and tools previously used to pursue an engagement in applied research is not presented.

(ii) Thus, the added value of Authors' proposal does not emerge. At the same time, the weakesnesses of previous approaches are not reported.

(iii) This, in turn, affects the clarity of Authors' rational. Authors presented a synthetic overview of the "compass" in figure 1. Having a quick look at this image reveals that their proposal consists in: a. analysis of need; b. generation of requirements; c. alpha prototype; d. beta-prototype. This look as a normally-used approach of participatory design where all stakeholders are involved in generation and evaluation of ideas of prototypes. What does Author's idea adds to existing approaches?

For instance, Authors relied on a business model as a theoretical framework. Fine...but why? how? Is it something replicable (I ask this because this choice stems from Authors' analysis of results and is not due to their analysis of existing solutions). Authors stated that they involved different researchers in different evaluation phases. This is fine, but why? Is it just a data-driven approach? To what extent can it be replicable and generalizable and useful compared to previous approaches?

Authors' title is "How to support a co-creative research approach in order to generate more impact". However, impact is not assessed because the real object of their research is not clearly stated.

I recommend Authors to work on their rational and avoid implicit discourse on the underpinnings of their approach. Otherwise, this would look just a mere post-hoc ad hoc analysis of their participatory design approach implementation in the healthcare field, without being a clear and useful information for other researchers or practitioners.

6. PLOS authors have the option to publish the peer review history of their article (what does this mean?). If published, this will include your full peer review and any attached files.

Reviewer #1: No

Reviewer #2: No

---

## [Author Response · Author response to Decision Letter 0]

10 Aug 2020

Dear Editor,

A rebuttal letter that responds to each point raised by the academic editor and reviewer has been uploaded as a separate file labeled 'Response to Reviewers'. We also included the marked-up copy of our manuscript that highlights changes made to the original version. It has been uploaded as a separate file labeled 'Revised Manuscript with Track Changes'.

We followed the guidelines for resubmitting our figure files and PLOS ONE's style requirements, including those for file naming. We also included captions for the Supporting Information files at the end of our manuscript, and updated any in-text citations to match accordingly. 

We hope that the manuscript is acceptable for publication in your journal in its present form.

---

## [Decision Letter · Decision Letter 1]

8 Sep 2020

PONE-D-20-05698R1

How to support a co-creative research approach in order to foster impact. The development of a Co-creation Impact Compass for healthcare researchers

PLOS ONE

Dear Dr. van Dijk-de Vries

Thank you for submitting your manuscript to PLOS ONE. After careful consideration of the revised version, we feel that it does meet PLOS ONE’s publication criteria. However, One of the Reviewer raised some minor issues.  Therefore, we invite you to submit a revised version of the manuscript that addresses the points raised during the review process.

Please submit your revised manuscript by 30th September. If you will need more time than this to complete your revisions, please reply to this message or contact the journal office at plosone@plos.org. Please include the following items when submitting your revised manuscript:

We look forward to receiving your revised manuscript.

Kind regards,

Paola Iannello

Academic Editor

PLOS ONE

Reviewers' comments:

Reviewer's Responses to Questions

**Comments to the Author**

1. If the authors have adequately addressed your comments raised in a previous round of review and you feel that this manuscript is now acceptable for publication, you may indicate that here to bypass the “Comments to the Author” section, enter your conflict of interest statement in the “Confidential to Editor” section, and submit your "Accept" recommendation.

Reviewer #1: All comments have been addressed

Reviewer #2: All comments have been addressed

2. Is the manuscript technically sound, and do the data support the conclusions?

Reviewer #1: Yes

Reviewer #2: Yes

3. Has the statistical analysis been performed appropriately and rigorously? 

Reviewer #1: N/A

Reviewer #2: N/A

4. Have the authors made all data underlying the findings in their manuscript fully available?

Reviewer #1: Yes

Reviewer #2: Yes

5. Is the manuscript presented in an intelligible fashion and written in standard English?

Reviewer #1: Yes

Reviewer #2: Yes

6. Review Comments to the Author

Reviewer #1: I appreciated all the shape changes made and in my opinion the paper is more linear and tidier in its structure. However, I still have two main concerns:

1) I would modify the abstract (27-33) by stating firstly how the instrument was created (i.e. by combining the business model with tools from design thinking) and only after pointing on the possibility that this tool (compassion) MIGHT help researchers to select valid co-creation tools which can help them to engage stakeholders, etc.

2) Linked to the first comments, given the importance that the tool might have for researchers, I wonder if it would not be appropriate to move the “supplementary material 2” directly into the article so that it can be directly explored (perhaps creating a single table) perhaps by briefly going into the description of the tools (which are a bit poor). If it were a problem with the number of images/tables I would recommend removing image 2 (maybe you can refer back to the original article since the model is now briefly explained in the text).

Reviewer #2: I appreciated Authors clarity and rigor in addressing all my concerns. Now, I feel the manuscript can be accepted.

7. PLOS authors have the option to publish the peer review history of their article (what does this mean?). If published, this will include your full peer review and any attached files.

Reviewer #1: No

Reviewer #2: No

---

## [Author Response · Author response to Decision Letter 1]

28 Sep 2020

Our rebuttal letter that responds to the points raised by reviewer 1 has been uploaded as a separate file labeled 'Response to Reviewers'.

---

## [Editor Report · Decision Letter 2]

29 Sep 2020

How to support a co-creative research approach in order to foster impact. The development of a Co-creation Impact Compass for healthcare researchers

PONE-D-20-05698R2

Dear Dr. Anneke van Dijk-de Vries, 

We’re pleased to inform you that your manuscript has been judged scientifically suitable for publication and will be formally accepted for publication once it meets all outstanding technical requirements.

Kind regards,

Paola Iannello

Academic Editor

PLOS ONE
---

## [Editor Report · Acceptance letter]

1 Oct 2020

PONE-D-20-05698R2 

How to support a co-creative research approach in order to foster impact. The development of a Co-creation Impact Compass for healthcare researchers 

Dear Dr. van Dijk-de Vries:

I'm pleased to inform you that your manuscript has been deemed suitable for publication in PLOS ONE. Congratulations! Your manuscript is now with our production department. 

Kind regards, 

on behalf of

Dr. Paola Iannello 

Academic Editor

PLOS ONE